# Geraniol-Mediated Suppression of Endoplasmic Reticulum Stress Protects against Cerebral Ischemia–Reperfusion Injury via the PERK-ATF4-CHOP Pathway

**DOI:** 10.3390/ijms24010544

**Published:** 2022-12-29

**Authors:** Yu Wu, Xiaomei Fan, Sha Chen, Ling Deng, Lu Jiang, Shaonan Yang, Zhi Dong

**Affiliations:** The Key Laboratory of Biochemistry and Molecular Pharmacology, College of Pharmacology, Chongqing Medical University, Chongqing 400010, China

**Keywords:** cerebral ischemia–reperfusion injury (CIRI), geraniol, endoplasmic reticulum (ER) stress, apoptosis

## Abstract

Endoplasmic reticulum (ER) stress plays an important role in cerebral ischemia–reperfusion injury (CIRI). Geraniol has antioxidant, antibacterial, and anti-inflammatory activities. Studies have shown that geraniol has a protective effect against CIRI in rats, but the exact mechanism is unclear. Purpose: The aim of this study was to investigate the protective mechanism of geraniol against CIRI. We established a middle cerebral artery occlusion reperfusion model in rats and a PC12 cell oxygen–glucose deprivation/reoxygenation (OGD/R) model to observe the neuroprotective effects of geraniol. Neurological scoring, 2,3,5-triphenyltetrazolium chloride staining, and hematoxylin and eosin staining were used to evaluate the neuroprotective effects of geraniol against CIRI. ER-stress-related and apoptosis-related protein expression was detected via Western blotting and immunofluorescence. Apoptosis was also detected via TUNEL assays and flow cytometry. The fluorescent detection of intracellular calcium was achieved using fluorescent calcium-binding dyes, and transmission electron microscopy was used to assess the neuronal ultrastructure. Geraniol effectively attenuated cerebral infarction and pathological injury after CIRI, had a protective effect against CIRI, significantly reduced the expression of the ER-stress-related proteins P-PERK, ATF4, CHOP, and GRP78 and the pro-apoptotic protein BAX, increased the expression of the anti-apoptotic protein BCL-2, and reduced the occurrence of apoptosis. In the OGD/R model in PC12 cells, the protective effect of geraniol was the same as that in vivo. Our results suggest that geraniol has a protective effect against ischemic stroke by a mechanism possibly related to ER stress via the PERK-ATF4-CHOP pathway.

## 1. Introduction

Stroke is one of the most common illnesses worldwide, with high mortality and morbidity rates, and 80% of all strokes are caused by embolism-induced ischemia [1]. Patients may suddenly experience paralysis, impaired speech, or loss of vision because of blood flow interruption (ischemia) caused by thrombosis or embolism [2]. Currently, the main treatment for ischemic stroke is thrombolysis or endovascular thrombectomy to rapidly restore blood flow to the brain [3]. However, because of the complex pathology of ischemic stroke, the treatment process to restore the blood supply to the brain may be accompanied by severe pathological damage to brain tissue, such as cerebral edema, brain hemorrhage, and neuronal death. This phenomenon is known as cerebral ischemia–reperfusion injury (CIRI) [4]. However, no effective clinical strategy exists for the prevention and treatment of CIRI. Thus, effective methods and drugs to treat CIRI must urgently be found.

The pathological process of CIRI is complex and unclear. Multiple mechanisms, such as free-radical-induced damage and inflammation, energy metabolism disruption in brain tissue, excitatory amino acid toxicity, calcium overload, oxidative stress damage, and apoptosis, may all be involved in CIRI [5,6]. Among them, apoptosis is an important part of CIRI and is critical in neurological function loss. Studies have shown that the inhibition of apoptosis can significantly improve CIRI. Therefore, attenuating apoptosis may be an effective method to treat CIRI [7,8].

The endoplasmic reticulum (ER) is a reticular membranous network that extends through the cytoplasm of the cell and is contiguous with the nuclear envelope. It can therefore sense and transmit signals that originate in any cellular subcompartment. To carry out the folding of intra-organellar, secretory, and transmembrane proteins in the ER, the ER has evolved as a specialized protein-folding machine that promotes productive folding and prevents aggregation [9,10]. ER homeostasis is critical for controlling various intracellular physiological functions including protein folding, calcium homeostasis, lipid metabolism, cell differentiation, and protein translocation [11,12,13]. However, under certain circumstances such as hypoxia, calcium flux and reactive oxygen species accumulation can disrupt ER homeostasis, resulting in unstructured ER protein aggregation and ER stress [14,15]. Such ER stress can, when mild, activate the unfolded protein response (UPR) in an effort to restore homeostasis [16,17]. Molecularly, the UPR can initiate several intracellular responses. First, the overall protein synthesis is attenuated while favoring the upregulation of protein chaperones to promote protein processing and refolding. In the event when increased chaperones cannot meet the folding/refolding demand, the UPR triggers protein catabolism via the ER-associated degradation pathway. The UPR mechanism also gears toward expanding the physical space of ER by increasing the synthesis of ER membranes through promoting the lipid metabolism. Finally, if these mechanisms fail to reverse chronic ER stress, the UPR will induce cell death via apoptosis. [13,17,18].

Activating transcription factor 6 (ATF6), double-stranded RNA-activated protein kinase-like ER kinase (PERK), and inositol-requiring enzyme 1 (IRE1) are three critical markers associated with the UPR. PERK, a transmembrane protein with an ER luminal stress-sensitive structural domain [19], first dissociates from GRP78 (78 kDa glucose regulated protein) during prolonged ER stress and then autophosphorylates, multimerizes, and phosphorylates the eukaryotic translation initiator factor (eIF2α) to activate the transcription of specific gene subsets, including ATF4 and CHOP [18,20]. Additionally, CHOP is an important indicator of ER-stress-induced apoptosis and can upregulate the Bax/BCL-2 ratio and caspase-3 cleavage [21,22].

Recently, ER stress has been associated with the physiopathological processes of many diseases, including tumors, diabetes, cardiovascular diseases, and autoimmune system diseases [23,24]. Numerous studies have confirmed that ER-stress-induced apoptosis plays a crucial role in CIRI [25].

Geraniol (trans-3,7-dimethyl-2,6-octadien-1-ol, chemical formula: C_10_H_18_O) is an important isoprenoid, the main ingredient of the oils of rose and palmarosa, and an active constituent in lemon, ginger, rose, and orange essential oils [26,27]. Geraniol has several biological activities, including antioxidant, antibacterial, anti-inflammatory, and antiangiogenic activity [26,28]. Additionally, geraniol has demonstrated a neuroprotective effect against CIRI and is involved in various signaling pathways of oxidative stress and apoptosis [29].

Considering these pharmacological effects, the aim of this study was to investigate the protective effect of geraniol during CIRI and its relationship with ER stress.

## 2. Results

### 2.1. Protective Effect of Geraniol against I/R-Induced Brain Damage

To investigate the neuroprotective effect of geraniol against I/R, we established a rat middle cerebral artery occlusion (MCAO) model and injected different geraniol concentrations for 5 consecutive days before surgery. After 24 h of reperfusion, the neurobehavioral score was higher in the I/R group than that in the sham group. The geraniol and edaravone groups improved neurological function in rats (Figure 1A). The TTC staining results showed no infarction in sham-operated rats. Additionally, cerebral infarction was visible in the I/R group. Compared with the I/R group, the cerebral infarct volume was significantly reduced in the I/R + geraniol group and was dose-related (Figure 1B,C). HE staining showed that compared with those in the sham-operated group, right cortical nerve cells in the I/R group exhibited obvious atrophy, breakage, absence, or nucleation, and brain tissue sections showed severe damage. However, the nuclei in the I/R + geraniol group were more intact, the number of atrophied nerve cells was significantly decreased, and brain damage was significantly lessened (Figure 1D). These results suggest that geraniol has a neuroprotective effect against I/R-induced brain damage. Geraniol at 200 mg/kg showed the best protective effect. So, we used this dose in subsequent experiments to conduct further mechanism research.

### 2.2. Identification of ER Stress Regulators via Metascape Online Analysis

To determine whether ER stress affects CIRI, we searched 59 ER-stress-regulated genes (Appendix A) from published papers and existing bioinformatics databases of our group, and the data sources are displayed in the Appendix A. These ER stress regulators were analyzed online using Metascape (https://metascape.org/gp/index, accessed on 24 April 2022). These results suggested that ER stress was involved in regulating biological processes, including GO:0043065: positive regulation of apoptotic processes; GO:2001236: regulation of the extrinsic apoptotic signaling pathway; GO:0043523: regulation of neuron apoptotic process; GO:0040007: growth, and so on (Figure 2A,B) (marked with red frame), which are connected to the pathological processes during CIRI. According to the results of network of enriched terms, it showed that ER stress was involved the process of the intrinsic apoptotic signaling pathway, the positive regulation of apoptotic process, the regulation of the extrinsic apoptotic signaling pathway, and the regulation of the neuron apoptotic process, which are connected to the pathological processes during CIRI (Figure 2C,D). These essential ER stress regulators interacted with multiple genes in a protein–protein interaction network (Figure 2E,F) (marked with red frame). The results suggested that ER stress was associated with apoptotic molecules such as Casp3, BCL-2, Bcl2l1, Bcl2l11, Bak1, Bag2, Bag5, Bag1, and Rnasel (marked with red frame). Based on the experimental results, we hypothesized whether the protective effect of geraniol after cerebral ischemia–reperfusion was related to endoplasmic reticulum stress.

### 2.3. Geraniol Reduces I/R-Induced ER Stress and Cell Apoptosis

Subsequently, we investigated whether geraniol could attenuate apoptosis after I/R in rats by inhibiting ER stress. The expression levels of GRP78, a signature protein of ER stress, CHOP, and BAX, a pro-apoptotic protein, were significantly higher in the I/R group than those in the sham-operated group and were dose-dependently reduced after geraniol treatment compared with those in the model group. Moreover, the expression level of the anti-apoptotic protein BCL-2 in the I/R group was significantly decreased compared with that in the sham-operated group and was dose-dependently increased after geraniol treatment compared with that in the I/R group (Figure 3A,B). Additionally, the immunofluorescence results also showed that CHOP expression was increased in the I/R group and decreased in the I/R + 200 mg/kg geraniol group (Figure 3C,D). To investigate the effect of geraniol on apoptosis after cerebral I/R in rats, brain slices were examined using an in situ end-labeling method (TUNEL). As shown in Figure 3E,F, TUNEL-positive cells were rare in the sham group, whereas substantial TUNEL-positive neurons were found in the I/R group. However, the number of TUNEL-positive cells in the I/R + 200 mg/kg geraniol group was significantly lower than that in the I/R group. These results suggest that geraniol attenuates I/R-induced apoptosis in rats by inhibiting ER stress.

### 2.4. Geraniol Regulates the ER-Stress-Mediated PERK-ATF4 Pathway in the Cerebral Cortex after I/R

The PERK-ATF4 pathway regulates CHOP expression during ER stress. Western blot experiments showed that the PERK expression did not change, and the P-PERK and ATF4 protein expression was significantly increased in the I/R group compared with that in the sham-operated group and significantly reduced in the I/R + 200 mg/kg geraniol group compared with that in the model group (Figure 4A,B). We also detected the P-PERK expression in the cerebral cortex via immunofluorescence. The P-PERK expression was significantly higher in the I/R group than that in the sham group and significantly lower in the I/R+200 mg/kg geraniol group than that in the I/R group (Figure 4C,D). Moreover, transmission electron microscopy indicated that the sham-operated group exhibited intact nuclear structures with smooth and continuous ER morphology, while the I/R group showed the most obvious neuronal damage with the absence of the nucleus pulposus, many vacuoles, and unclear or absent ER. Compared with the I/R group, the I/R+200 mg/kg geraniol group exhibited reduced neuronal damage, improved the nuclear structure, and allowed fewer vacuoles to form (Figure 4E). These results suggested that geraniol inhibited the PERK-ATF4 pathway in the cerebral cortex after I/R.

### 2.5. Inhibition of ER Stress via the PERK-ATF4-CHOP Pathway Is Required for the Geraniol-Mediated Protective Effect against I/R in the Cerebral Cortex

To explore the molecular mechanism of geraniol-mediated protection against I/R in the cerebral cortex, we injected the PERK pathway agonist CCT020312 via the lateral ventricle preoperatively (Figure 5A). The I/R + geraniol + CCT020312 group showed higher mNSS scores than those in the I/R + 200 mg/kg geraniol group (Figure 5B). TTC staining showed a significant increase in brain infarct volume in the I/R + geraniol + CCT020312 group compared with that in the I/R + 200 mg/kg geraniol group (Figure 5C,D). Brain tissue sections in the I/R + geraniol + CCT020312 group showed severe damage compared with those in the I/R + 200 mg/kg geraniol group (Figure 5E).

Additionally, the PERK expression did not change, the P-PERK, ATF4, CHOP, GRP78, and BAX protein expression was significantly increased, and the BCL-2 protein expression was decreased in the I/R + geraniol + CCT020312 group (Figure 6A,B). We also detected P-PERK and CHOP expression in the rat cerebral cortex via immunofluorescence. The P-PERK and CHOP expression was significantly higher in the cerebral cortex of the I/R + geraniol + CCT020312 group than that in the I/R + 200 mg/kg geraniol group (Figure 6C–F). We also further verified the antiapoptotic effect of geraniol using a TUNEL assay. Compared with that in the I/R + geraniol group, positive TUNEL staining was significantly increased in the I/R + geraniol + CCT020312 group (Figure 6G,H). Similarly, transmission electron microscopy showed that the I/R + geraniol + CCT020312 group exhibited more severe neuronal damage, with nuclear membrane lysis, nuclear chromatin overflow, fusion of the ER with the cytosol, and the formation of vacuoles, than that in the I/R + 200 mg/kg geraniol group (Figure 6I). These results suggest that geraniol has a protective effect against CIRI by inhibiting ER stress via the PERK-ATF4-CHOP pathway.

### 2.6. Geraniol Increases Cell Viability of PC12 Cells Induced by OGD/R

To select the appropriate concentration of geraniol, we first treated PC12 cells with different geraniol concentrations under normal conditions. A CCK-8 assay showed that 20 μM of geraniol had no significant effect on the PC12 cell viability (Figure 7A). Subsequently, a reoxygenation time of 24 h was selected using the CCK-8 method (Figure 7B). The cell viability in the OGD/R + 20 μM geraniol group was significantly higher than that in the OGD/R control group at different times of reoxygenation at 6, 12, and 24 h. Additionally, in both the OGD/R control group and the OGD/R + geraniol group, a 24 h reoxygenation time increased cell viability. The cell morphology of PC12 cells after induction by OGD/R was observed under a microscope. The cells in the control group were in good condition, cells with prominent stems and branches, forming a dense network, while the dead cells in the OGD/R group increased, lost their normal shape, and presented a wider intercellular space. Compared with the OGD/R group, the number of cells in the geraniol-treated group increased, and the cells exhibited extensive branching and intact intercellular joints (Figure 7C). These results suggest that geraniol has a protective effect against OGD/R in PC12 cells.

### 2.7. Geraniol Reduces ER Stress and Cell Apoptosis in OGD/R-Induced PC12 Cells

For in vivo MCAO models, our experimental results demonstrated that geraniol attenuated CIRI by inhibiting ER-stress-dependent apoptosis. To verify the in vivo results, we exposed cultured PC12 cells to OGD/R. Western blot analysis was used to investigate ER-stress-associated protein expression levels. The OGD/R group exhibited significantly increased GRP78, CHOP, and BAX protein expression and decreased BCL-2 expression compared with the control group, while the OGD/R + 20 μM geraniol group had significantly decreased GRP78, CHOP, and BAX protein expression and increased BCL-2 expression compared with the OGD/R group (Figure 8A,B). Additionally, we detected apoptosis via flow cytometry, and compared with that in the control group, apoptosis was increased in the OGD/R group. However, apoptosis was significantly decreased in the OGD/R + 20 μM geraniol group compared with that in the OGD/R group (Figure 8C,D). These results indicated that geraniol significantly reduced ER stress and apoptosis in OGD/R-induced PC12 cells.

Subsequently, the mechanism of action of geraniol on OGD/R-induced PC12 cells was investigated using the PERK activator CCT020312 (Figure 9A). We first screened the appropriate CCT020312 concentration using a CCK-8 assay. The cells were divided into control, OGD/R, OGD/R+20μM geraniol, OGD/R+CCT020312, and OGD/R+geraniol+CCT020312 groups. The CCK-8 assay showed a significant decrease in cell survival after the simultaneous administration of CCT020312 and geraniol compared with that after OGD/R+20μM geraniol treatment (Figure 9B–C). The cell morphology of each group after OGD/R induction was also observed via microscopy (Figure 9D). Compared with the OGD/R+20μM geraniol group, the OGD/R+geraniol+CCTO20312 group had more dead cells, wider intercellular spaces, and less obvious cell branching. Consistent with the in vivo results, the PERK expression did not change, the P-PERK, ATF4, CHOP, GRP78, and BAX expression levels were significantly increased, and BCL-2 expression was decreased in the OGD/R+geraniol+CCTO20312 group compared with that in the OGD/R+20μM geraniol group (Figure 10A,B). We also detected P-PERK and CHOP expression in cells after OGD/R via immunofluorescence. The intracellular P-PERK and CHOP expression was significantly higher in the OGD/R group than that in the control group, was significantly reduced in the OGD/R+20μM geraniol group compared with that in the OGD/R group, and was significantly increased in the OGD/R+geraniol+CCT020312 group compared with the OGD/R+20μM geraniol group (Figure 10C–F). The flow cytometry results also showed that apoptosis was increased in the OGD/R+geraniol+CCT020312 group compared with that in the OGD/R+20μM geraniol group (Figure 10G–H).

Changes in the intracellular calcium ion concentration can reflect the ER stress intensity. After incubation with the fluorescent calcium probe Fluo-8/AM, changes in the fluorescence intensity of calcium ions in each cell group were observed via laser confocal microscopy, and the intracellular calcium ion concentration in each group was measured via flow cytometry. Compared with that in the control group, the fluorescence intensity was increased in the OGD/R group, i.e., the intracellular calcium ions were increased, while the intracellular fluorescence intensity was decreased in the OGD/R+20 μM geraniol group compared with that in the OGD/R group, i.e., calcium ions were decreased. The OGD/R + geraniol + CCT020312 group exhibited significantly increased fluorescence intensity, i.e., an increased calcium ion concentration, compared with the OGD/R + 20 μM geraniol group (Figure 10I–K). In addition, the cell transmission electron microscopy results showed that the control group had clear and intact cell structures, while the OGD/R group had severe damage, such as the condensation of chromatin in the inner nuclear membrane and obvious apoptosis. Compared with the OGD/R group, the OGD/R+20 μM geraniol group had more intact cell structures and less obvious nuclear damage, and compared with the OGD/R+20 μM geraniol group, cells in the OGD/R+geraniol+CCT020312 group exhibited cytoplasmic swelling and nuclear shrinkage (Figure 10L). All of the results showed that geraniol reduces apoptosis in OGD/R-treated PC12 cells by inhibiting ER stress via the PERK-ATF4-CHOP pathway.

## 3. Discussion

Ischemic stroke, a complicated and devastating disease, is one of the top three causes of death worldwide following cancer and heart disease [30,31]. The effective treatment of neurological damage caused by ischemic stroke remains a major challenge in clinical research. In recent years, many studies have examined the therapeutic effects of geraniol on diseases such as liver cancer, breast cancer, colon cancer, neuropathic pain, and depression. Previous studies confirmed that geraniol has a protective effect against cerebral I/R, but the exact mechanism is unclear [32]. In this study, we performed MCAO in rats to establish a transient CIRI model, and geraniol was injected for 5 consecutive days before surgery. Geraniol significantly reduced infarct volume and improved neurological dysfunction, which indicated a protective effect of geraniol against I/R.

CIRI results in cell dysfunction, injury, or death, which triggers calcium overload, oxidative stress, organelle dysfunction, metabolic disorders, and inflammatory responses. ER stress is involved in the pathophysiological process of CIRI [33,34]. In the early stages of cerebral ischemia, mild ER stress promotes cell survival by activating the UPR and increasing the expression of quasi-survival proteins. Misfolded proteins within the ER can be degraded by cell membrane 26S proteases and lysosome-associated autophagy [35]. However, if ER stress is prolonged, high levels of toxic products are released, which can lead to cytoplasmic calcium overload and ultimately exacerbate ischemic neuronal damage and cause apoptosis [36,37]. According to our online analysis of 59 ER stress-related molecules using Metascape (https://metascape.org/gp/index, accessed on 24 April 2022), the results showed that these molecules are involved in biological processes such as the positive regulation of apoptotic processes, the regulation of extrinsic apoptotic signaling pathways, and the regulation of neuronal apoptotic processes and growth, and these are all related to pathological processes during CIRI. Therefore, CIRI can be reduced by inhibiting ER stress [38,39]. We detected the protein levels of the ER stress markers GRP78 and CHOP in both MCAO rat models and OGD/R-induced PC12 cells, and geraniol significantly inhibited GRP78 and CHOP expression. Therefore, we speculated that the protective effect of geraniol may be related to the inhibition of ER stress.

ER stress is triggered by the accumulation of unfolded or misfolded proteins in the ER [40]. The UPR exerts its regulatory role primarily through the activation of three transmembrane proteins: PERK, ATF6, and IRE1 [41]. Under physiological conditions, IRE-1, PERK, and ATF-6 are inhibited by binding to GRP78, an ER chaperone, also known as binding immunoglobulin and heat shock protein 5A [42]. Under conditions associated with ER dysfunction, GRP78 dissociates from these ER transmembrane proteins, initiating the UPR, a protective response to restore ER function. When stress persists or is excessive, the UPR fails to restore ER homeostasis and then activates the apoptotic signaling pathway [43,44].

When misfolded proteins accumulate in the ER lumen, PERK mediates the inhibition of translation through the phosphorylation of eIF2α, which efficiently eliminates the ER load. PERK-initiated eIF2α phosphorylation also contributes to the up-regulation of ATF4 [45]. ATF4 is a key player in several stress-response pathways and induces the expression of UPR-associated inflammatory signaling molecules, ER chaperones and trafficking machinery, antioxidative stress responses, and autophagy [46]. A downstream target of ATF4 is CHOP, which is an important transcription factor in the response to ER stress-mediated apoptosis and can be regulated by all three UPR signaling branches [47]. However, CHOP serves as a pivotal stimulus for cell death and is mainly induced by ATF4 [48]. Therefore, PERK, which is indicative of ER stress and UPR activation, is indispensable for translational regulation in ER-stressed cells, and the PERK-ATF4-CHOP pathway plays an important role in ER-stress-induced apoptosis.

To explore the specific protective mechanism of geraniol, we used CCT020312 in our experiments. CCT020312 is an agonist of PERK [49,50,51]. When CCT020312 was added, the phosphorylation of PERK was promoted, and then the expression level of P-PERK was increased, which in turn led to the increase in the levels of downstream molecules such as ATF4 and CHOP, and aggravated ER stress. We performed Western blotting to detect PERK-ATF4-CHOP pathway protein expression both in the MCAO rat model and OGD/R PC12 cells. Geraniol significantly decreased P-PERK and ATF4 expression. Additionally, TUNEL staining and flow cytometry indicated that geraniol significantly reduced apoptosis.

In summary, we showed that geraniol can reduce ER-stress-induced apoptosis after CIRI, and the protective mechanism may be achieved through the PERK-ATF4-CHOP signaling pathway (Figure 11). Our findings provide new ideas for the treatment of CIRI and have important clinical implications.

However, our study has some limitations. We demonstrated that geraniol may alleviate ER-stress-induced CIRI by downregulating the PERK-ATF4-CHOP pathway, but we did not investigate the relationship between geraniol and the two other ER stress pathways. Moreover, the pathological process of CIRI involves multiple complex mechanisms, such as oxidative stress, inflammation, and autophagy, and we did not investigate whether geraniol is related to these pathological processes. These topics will be investigated in future studies.

## 4. Materials and Methods

### 4.1. Animals

Male Sprague–Dawley rats (250–300 g) were used in this study. All rats were purchased from the Animal Center of Chongqing Medical University. The rats were kept at room temperature (21–25 °C) with a 12 h light–dark cycle and allowed to move freely with free access to food and water. All animal experiments were approved by the Animal Care and Use Committee of Chongqing Medical University.

### 4.2. Grouping and Drug Treatment

The rats were randomly divided into six groups to determine the appropriate dose of geraniol (98%, Sigma-Aldrich, St. Louis, MO, USA, 106-24-1) for subsequent analyses: the sham-operated group, the sham-operated with 200 mg/kg geraniol group, the middle cerebral artery occlusion (MCAO) group, the 50 mg/kg geraniol group, the 100 mg/kg geraniol group, the 200 mg/kg geraniol group, and the edaravone group (as a positive control, 3 mg/kg, intravenous). The rats were given geraniol via continuous intraperitoneal injection 5 days before MCAO surgery, and once again after surgery. Based on the results of the preceding experiment, the most appropriate dose of geraniol was chosen for the following experiment. The rats were randomly assigned to six groups: the sham-operated group, the ischemia–reperfusion (I/R) group, the I/R + 200 mg/kg geraniol group, the I/R + 5 mmol CCT020312 (PERK Activator, MCE, New Jersey, USA, HY-119240) group, and the I/R + 200 mg/kg geraniol+5 mmol CCT020312 group. The lateral ventricular injection of 10 μL of 5 mmol CCT020312 was performed 1.5 h prior to the MCAO operation. After 24 h of reperfusion, all rats were sacrificed, and brain tissue was harvested immediately. The same solvent was given to the sham-operated and I/R groups.

### 4.3. MCAO Model

Rats were anesthetized with 1% pentobarbital sodium and fixed in a supine position. The right common carotid artery, external carotid artery, and internal carotid artery were carefully separated, the proximal ends of the common and external carotid arteries were ligated, and the internal carotid artery was clamped with an arterial clip. A 2 cm oblique incision was made with vascular scissors at the bifurcation of the external and internal carotid arteries, and a monofilament was advanced into the internal carotid artery to a depth of 18.5 ± 1.0 mm until slight resistance was felt. At 1.5 h after occlusion, the nylon suture was withdrawn to allow for reperfusion, the wounds were sutured, and perfusion occurred for 24 h. The sham-operated rats underwent the same procedure, but the nylon monofilament was not inserted. Rats were housed in clean grade cages at 36–37 °C.

### 4.4. Neurological Function Evaluation

Neurological deficits and the cerebral infarction volume were used to assess the degree of ischemic injury. At 24 h after I/R, blinded observers scored the neurological deficits according to the modified neurological severity score (mNSS). The score was graded on a scale of 0 to 14 (normal score, 0; maximum score, 14). A score of 10 to 14 indicated severe injury; 5 to 9 moderate injury; and 1 to 4 mild injury.

### 4.5. 2,3,5-Triphenyltetrazolium Chloride (TTC) Staining of Brain Infarct Volume

TTC-stained brain sections were used to assess brain infarct volume. After 24 h of cerebral I/R, the rats were anesthetized and euthanized. Brains were removed and placed at −20 °C for 20 min. The brain tissue was fixed in the brain slot (Lab Anim Tech, Beijing, China, 14-0110) and cut into five coronal slices approximately 2 mm thick with a scalpel blade (Beyotime, Shanghai, China, FS205) on ice. Samples were incubated in 2% TTC for 15 min at 37 °C and fixed overnight in 4% paraformaldehyde, after which, they were removed and photographed. The white area of the brain indicated infarcted tissue, and the red area indicated normal tissue. The area of infarction in each slice was measured in Image J software, version 1.46 (NIH, Bethesda, MD, USA). The infarct areas of each section were obtained as the average of the sum of two sides. The volume of infarction for each animal was calculated by multiplying the average slice thickness (2 mm) by the sum of the infarct areas in all brain slices. The results are represented as a percentage of the total volume.

### 4.6. Hematoxylin and Eosin (HE) Staining

After 24 h of reperfusion, anesthetized rats were slowly perfused with phosphate-buffered saline (PBS) and 4% paraformaldehyde (4% PFA, Beyotime, Shanghai, China, P0099) via the left apical. The intact brain tissues were fixed for 24 h and then dehydrated with an alcohol gradient and paraffin embedded. Coronal sections (approximately 5 μm) were cut, and HE staining was performed to observe the histomorphology.

### 4.7. TUNEL Staining

The apoptosis of cortical cells in the rat penumbra was detected via TUNEL staining. Each section was observed and imaged using a light microscope. Normal cells had blue nuclei, and apoptotic cells with green nuclei were considered TUNEL-positive. The number of apoptotic cells in the cortical penumbra was calculated, and the number of apoptotic cells in each area was counted. The TUNEL-positive cell ratio was analyzed using Image J software, version 1.46.

### 4.8. Cell Culture

PC12 cells were purchased from ATCC (American Type Culture Collection, Manassas, VA, USA). The third generation was used for experiments. PC12 cells were cultured in standard growth medium (89% Dulbecco’s Modified Eagle Medium, DMEM, Gibico, CA, USA, 11965118), supplemented with 5% fetal bovine serum (FBS, 10% *v*/*v*; BIOAGRIO, Nanjing, China, S1356), 1% penicillin–streptomycin (Beyotime, Shanghai, China, 10,378–016), and 5% horse serum (Solarbio, Beijing, China, S9050) at 37 °C in a 5% CO_2_ atmosphere.

### 4.9. Oxygen–Glucose Deprivation/Reoxygenation (OGD/R)

Cells with suitable density and conditions were selected, and the complete medium was discarded. Cells were washed three times with PBS. Glucose-free DMEM (Gibico, CA, USA, 11966025) was added to simulate a lack of sugar, and the cells were incubated with 5% CO_2_, 94% N_2_, and 1% O_2_ to simulate hypoxia. After 2 h, the cells were cultured according to the original culture conditions to simulate reoxygenation and reglycation. The reoxygenation time was selected according to the experimental requirements.

### 4.10. Immunofluorescence Staining

Intact brain tissues were fixed for 24 h, dehydrated with an alcohol gradient, and paraffin embedded. Coronal sections (approximately 5 μm) were cut, and CHOP and P-PERK expression in the semi-dark zone of the cortical region was detected via immunofluorescence.

After interventions were applied, the cells were washed twice with PBS and fixed with 4% paraformaldehyde for 10 min. Then, 0.5% Triton X-100 (PBS preparation) was added for permeabilization at room temperature for 20 min. Normal goat serum was added dropwise, and cells were incubated at room temperature for 30 min. A sufficient amount of diluted primary antibody was added dropwise, and cells were incubated overnight at 4 °C. The primary antibody was recovered, a fluorescent secondary antibody was added, and cells were incubated for 1 h at room temperature. Then, DAPI (Beyotime, Shanghai, China, C1002) was added dropwise, and cells were incubated for 10 min at room temperature in the dark. The specimen was stained for nucleation and observed and imaged with a fluorescence microscope (Olympus FV1000, Tokyo, Japan) using Alexa Fluor 350 excitation/emission wavelengths at 346 nm/422 nm.

### 4.11. Transmission Electron Microscopy

After the rats were anesthetized, PBS was injected into the left ventricle before fixation with 2.5% glutaraldehyde in paraformaldehyde. Fresh brain tissue was immediately removed and cut into ultrathin cortical sections, which were immersed in glutaraldehyde. PC12 cells were collected after 24 h of OGD/R, digested, and centrifuged, and the supernatant was discarded. Glutaraldehyde was added to fix the cells. Changes in the neuronal ultrastructure were observed via transmission electron microscopy.

### 4.12. Western Blotting

Rat brains were removed, and the cerebral cortex was isolated for subsequent experiments. PC12 cells were collected after 24 h of OGD/R, lysed with RIPA buffer (Beyotime, Shanghai, China, P0013K), and centrifuged, and the supernatant was collected, mixed with loading buffer, and stored at −20 °C. Then, 40 μg/lane proteins were loaded onto 10% sodium dodecyl sulfate–polyacrylamide gels (Epizyme, Shanghai, China) and transferred to a nitrocellulose filter membrane. The membrane was washed with tris-buffered saline and 1% Tween 20 (Beyotime, Shanghai, China, ST673) and blocked with 5% non-fat milk for 2 h to eliminate non-specific protein binding. Next, the membrane was incubated with the following primary antibodies at 4 °C: PERK (dilution: 1:1000, Affinity, OH, USA, AF5304), P-PERK (dilution: 1:1000, Affinity, OH, USA, AF4499), ATF-4 (dilution: 1:800, Affinity, OH, USA, DF6008), GRP78 (dilution: 1:1000, Bioss, Beijing, China, bs-1219R), BAX (dilution: 1:1000, Affinity, OH, USA, AF0120), BCL-2 (dilution: 1:1000, Affinity, OH, USA, AF6139), CHOP (dilution: 1:1000, Affinity, OH, USA, AF6277), and β-actin (dilution: 1:1000, Proteintech, Wuhan, China, 10021293). Finally, the membrane was incubated with the following secondary antibodies: peroxidase-conjugated goat anti-mouse IgG (H + L) (Proteintech, Wuhan, China, SA00006-2) and peroxidase-conjugated goat anti-rabbit IgG (H + L) (Proteintech, Wuhan, China, B900210). A Chemidoc XRS system was used to capture digital images of the protein bands.

### 4.13. Cell Viability Assay

Cells were inoculated into 96-well plates and treated according to experimental conditions, and then, 10 µL of Cell Counting Kit-8 (CCK-8, Beyotime, Shanghai, China, C0037) solution was added to each well. The cells were incubated at 37 °C for 30 min, and the absorbance was measured at 450 nm.

### 4.14. Flow Cytometric Apoptosis Assays

After cell interventions, PC12 cells were stained with Annexin V and 7-AAD (Solarbio, Beijing, China, CA1030) and then analyzed via flow cytometry to quantify apoptosis. Briefly, after OGD/R, both suspended and adherent PC12 cells were harvested and washed with cold PBS. Next, the cells were resuspended in a working solution containing Annexin V-PE and 7-AAD and incubated at room temperature for 15 min in the dark. PC12 cells were immediately analyzed via flow cytometry. According to the quadrant diagram of the flow cytometry results, the lower right quadrant was the early apoptotic cell, and the upper right quadrant was the late apoptotic cell. The sum of the two quadrant values was the apoptosis cell ratio. More than 1 × 10^4^ cells were recorded in each sample, and the experiment was repeated three times.

### 4.15. Intracellular Calcium Ion Measurement

PC12 cells were divided into five groups: control, OGD/R, OGD/R + geraniol, OGD/R + CCT020312, and OGD/R + geraniol + CCT020312. Fluo-8/AM (AAT Bioquest, California, USA, 1345980-40-6) storage solution (10 μL) and 10 μL of Pluronic F-127 storage solution were diluted to 3 mL with HHBS (Hanks’Buffer with 20 mM Hepes, AAT Bioquest, California, USA, AAT-20011). Then, 1 mL of the mixture was added to each cell sample. After incubation at 37 °C and 5% CO_2_ for 30 min, the probes were washed with HHBS, intracellular calcium was detected using fluorescent calcium-binding dyes, and fluorescence changes in the dye-bound calcium were detected via confocal laser microscopy and flow cytometry. Experimental conditions: excitation wavelength: 488 nm; emission wavelength: 506 nm.

### 4.16. Statistical Analysis

All values are expressed as means ± standard deviation. Comparisons among multiple groups were performed with one-way analysis of variance. When the differences were considered statistically significant, the least-significant difference test was performed for comparisons between two groups. Statistical analysis was performed using SPSS 17.0 software. A value of *p* < 0.05 was considered statistically significant.

## 5. Conclusions

In summary, our study demonstrates that geraniol reduces neural injury induced by cerebral I/R and the occurrence of apoptosis. The underlying mechanism may be related to the ER-stress-associated PERK-ATF4-CHOP signaling pathway.

## Figures and Tables

**Figure 1 ijms-24-00544-f001:**
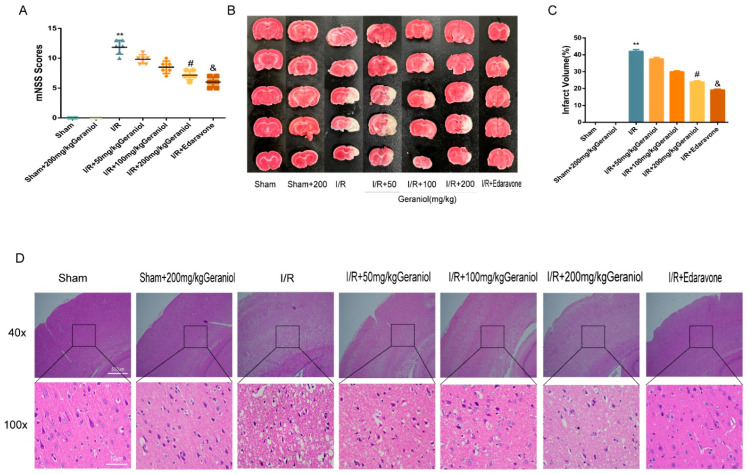
Protective effect of geraniol against ischemia–reperfusion (I/R)-induced brain damage. (**A**) Modified neuronal severity score (mNSS) of ischemic rats and score analysis (*n* = 6). (**B**,**C**) Representative 2, 3, 5-triphenyl tetrazolium chloride staining images and statistical analysis of the brain infarct volume (*n* = 3). (**D**) Representative hematoxylin-and-eosin-stained cerebral slices were imaged via microscopy. ** *p* < 0.01 I/R group vs. sham group, # *p* < 0.01 I/R + 200 mg/kg geraniol group vs. I/R group, & *p* < 0.01 I/R + edaravone group vs. I/R group.

**Figure 2 ijms-24-00544-f002:**
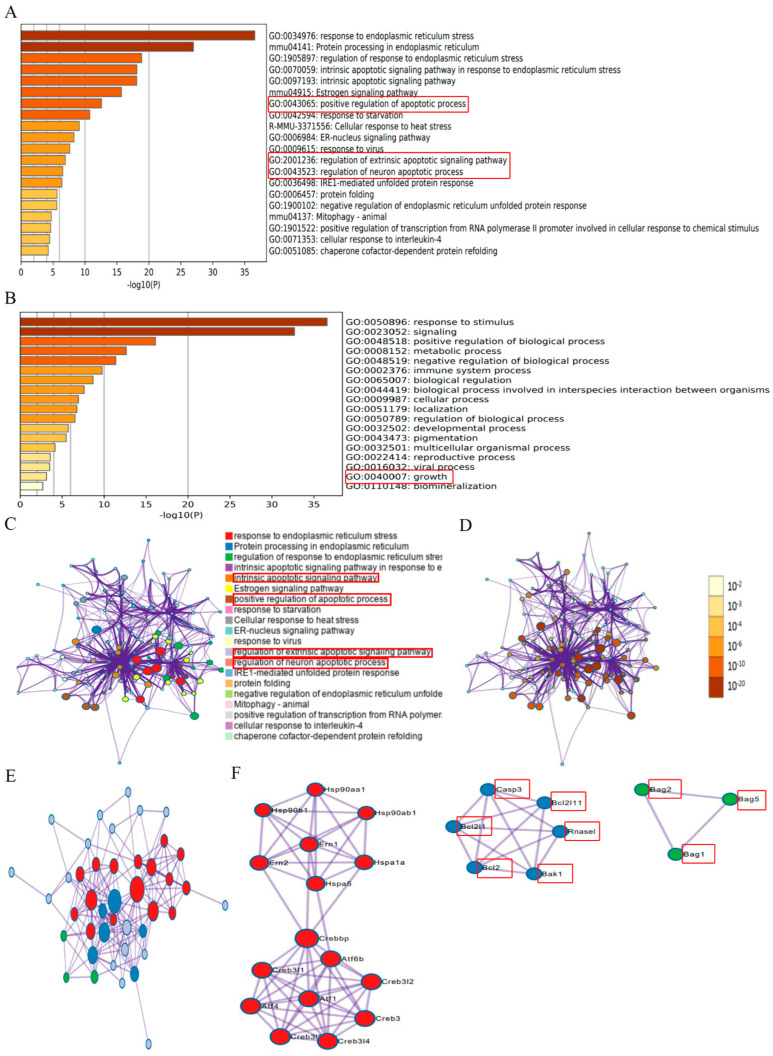
Identification of endoplasmic reticulum (ER) stress regulators via Metascape online analysis. (**A**) Bar graph of enriched terms across ER stress gene lists; colors indicate *p*-values. (**B**) The top-level Gene Ontology biological processes are shown. (**C**) Network of enriched terms: color-coded by cluster ID, where nodes that share the same cluster ID are typically close to each other. (**D**) Network colored-coded by *p*-value, where terms containing more genes tend to have a more significant *p*-value. (**E**,**F**) Protein–protein interaction network and MCODE components identified in the ER stress gene lists.

**Figure 3 ijms-24-00544-f003:**
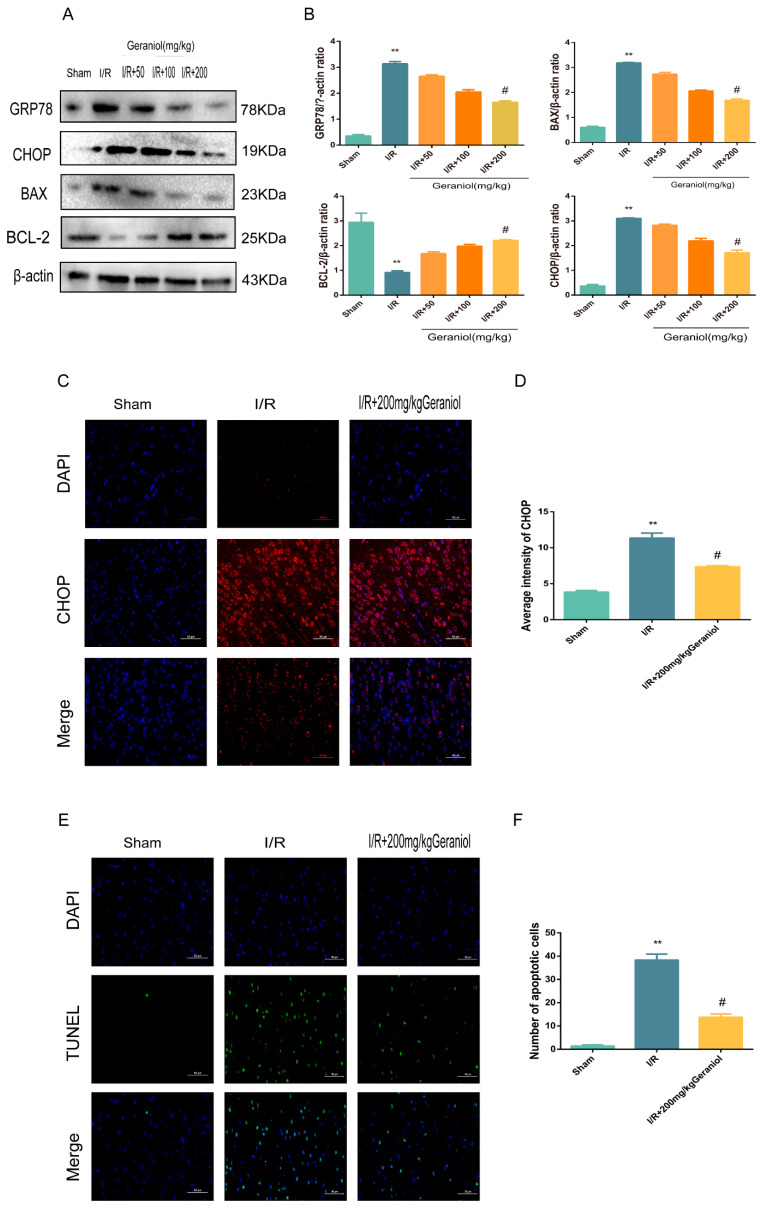
Geraniol reduces ischemia–reperfusion (I/R)-induced endoplasmic reticulum stress and cell apoptosis. (**A**,**B**) Western blot analysis of GRP78, BAX, BCL-2, and CHOP expression in the cortex (*n* = 6). (**C**,**D**) Immunofluorescence staining of CHOP in the cortex. Representative images were acquired under 400× magnification, scale bars = 50 µm. (**E**,**F**) Apoptosis in the cortex detected via TUNEL assays (*n* = 3). Representative images were acquired under 400× magnification, scale bars = 50 µm. ** *p* < 0.01 I/R group vs. sham group, # *p* < 0.01 I/R+200 mg/kg geraniol group vs. I/R group.

**Figure 4 ijms-24-00544-f004:**
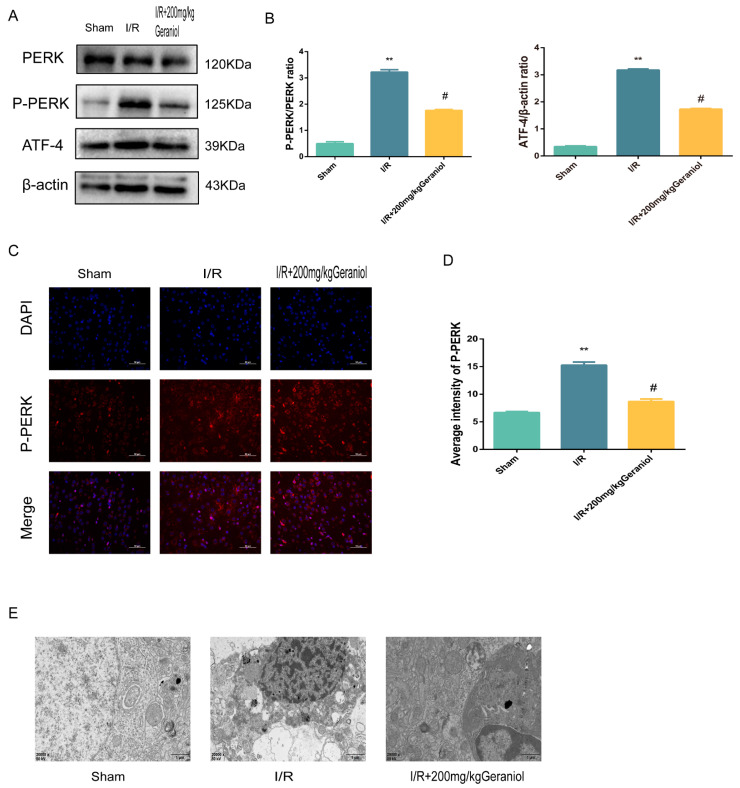
Geraniol regulates induction of the endoplasmic-reticulum-stress-mediated PERK-ATF4 pathway by ischemia–reperfusion (I/R) in the cerebral cortex. (**A**,**B**) Western blot analysis of PERK, P-PERK, and ATF4 expression in the cortex (*n* = 6). (**C**,**D**) Immunofluorescence staining of P-PERK in the cortex (*n* = 3). Representative images were acquired under 400× magnification, scale bars = 50 µm. (**E**) Electron microscopy images of neuronal nuclei in the cerebral cortex of rats (20,000×). ** *p* < 0.01 I/R group vs. sham group, # *p* < 0.01 I/R + 200 mg/kg geraniol group vs. I/R group.

**Figure 5 ijms-24-00544-f005:**
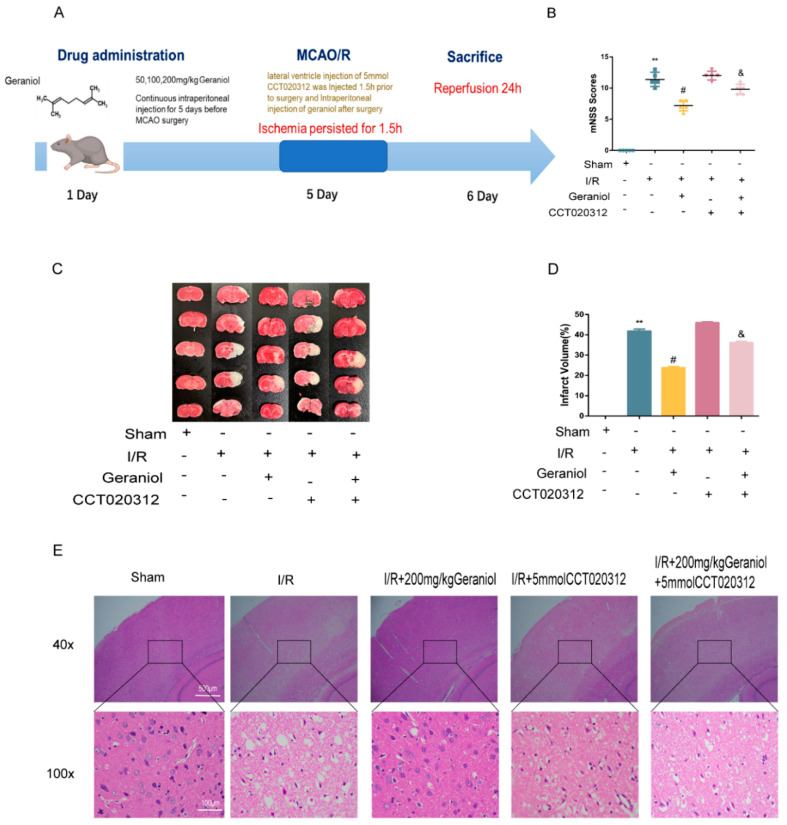
Geraniol had a protective effect on cerebral cortex ischemia–reperfusion (I/R) through the PERK-ATF4-CHOP signaling pathway in vivo. (**A**) Experimental protocol for rats. (**B**) Modified neurological severity score (mNSS) of ischemic rats and score analysis (*n* = 6). (**C**,**D**) Representative images of 2, 3, 5-triphenyl tetrazolium chloride staining and statistical analysis of the brain infarct volume *(n* = 3). (**E**) Representative hematoxylin-and-eosin-stained cerebral slices imaged via microscopy. ** *p* < 0.01 I/R group vs. sham group, # *p* < 0.01 I/R + 200 mg/kg geraniol group vs. I/R group, & *p* < 0.01 I/R + 200 mg/kg geraniol+5 mmol CCT020312 group vs. I/R + 200 mg/kg geraniol group.

**Figure 6 ijms-24-00544-f006:**
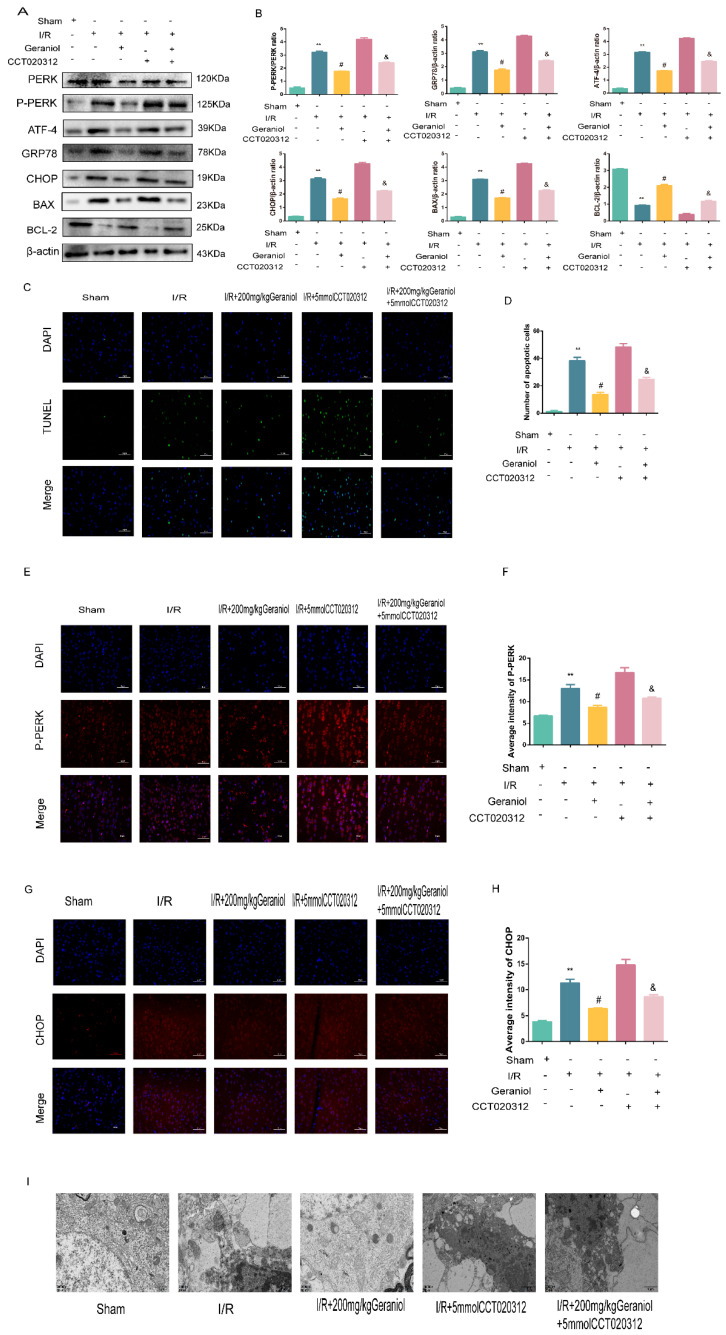
Geraniol-mediated protection against CIRI requires inhibition of endoplasmic reticulum stress via the PERK-ATF4-CHOP pathway. (**A**,**B**) Western blot analysis of PERK, P-PERK, ATF4, GRP78, BAX, BCL-2, and CHOP expression in the cortex (*n* = 6). (**C**–**F**) Immunofluorescence staining of P-PERK and CHOP in the cortex. Representative images were acquired under 400× magnification, scale bars = 50 µm (*n* = 3). (**G**,**H**) Apoptosis was detected in the cortex by TUNEL assays (*n* = 3). Representative images were acquired under 400× magnification, scale bars = 50 µm. (**I**) Electron microscopy images of neuronal nuclei in the rat cerebral cortex (20,000×). ** *p* < 0.01 I/R group vs. sham group, # *p* < 0.01 I/R + 200 mg/kg geraniol group vs. I/R group, & *p* < 0.01 I/R + 200 mg/kg geraniol+5 mmol CCT020312 vs. I/R + 200 mg/kg geraniol group.

**Figure 7 ijms-24-00544-f007:**
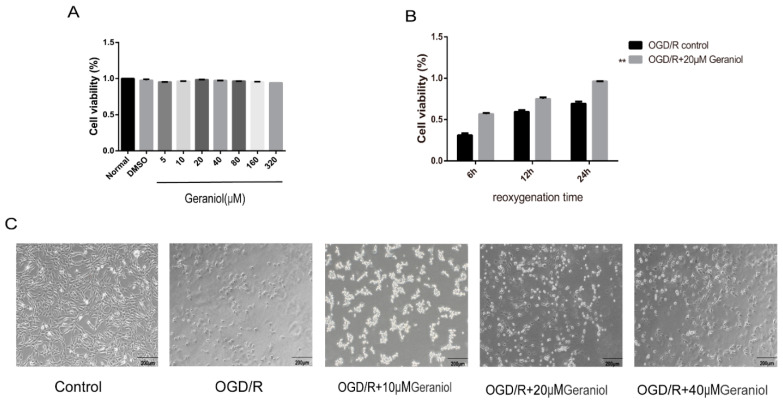
Geraniol increases the cellular activity of oxygen–glucose deprivation/reoxygenation (OGD/R)-induced PC12 cells. (**A**,**B**) Cell viability was assayed using the CCK-8 kit (*n* = 3). (**C**) Microscopic observation of OGD/R-treated PC12 cell morphology. (**B**) ** *p* < 0.01 OGD/R + 20 μM geraniol group vs. OGD/R control group.

**Figure 8 ijms-24-00544-f008:**
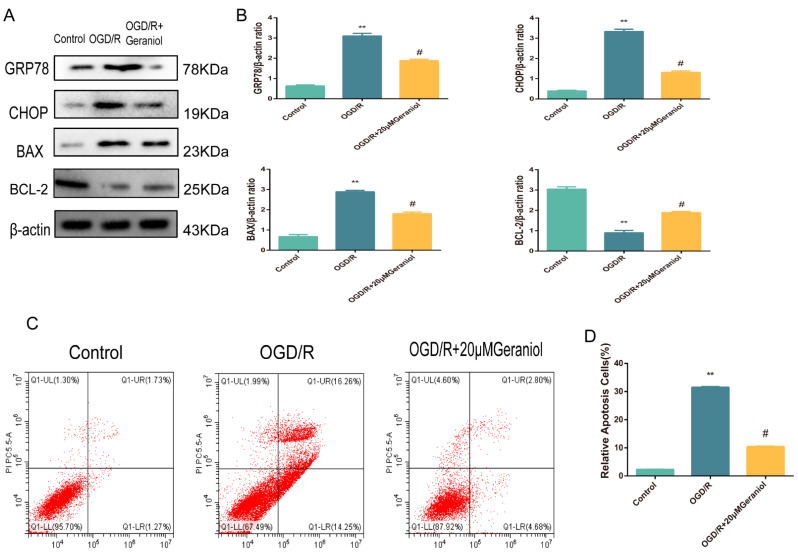
Geraniol reduces endoplasmic reticulum stress and cell apoptosis in oxygen–glucose deprivation/reoxygenation (OGD/R)-induced PC12 cells. (**A**,**B**) Western blot analysis of GRP78, BAX, BCL-2, and CHOP expression. (**C**,**D**) Apoptosis detection via flow cytometry. ** *p* < 0.01 OGD/R group vs. control group, # *p* < 0.01 OGD/R + 20 μM geraniol group vs. OGD/R group.

**Figure 9 ijms-24-00544-f009:**
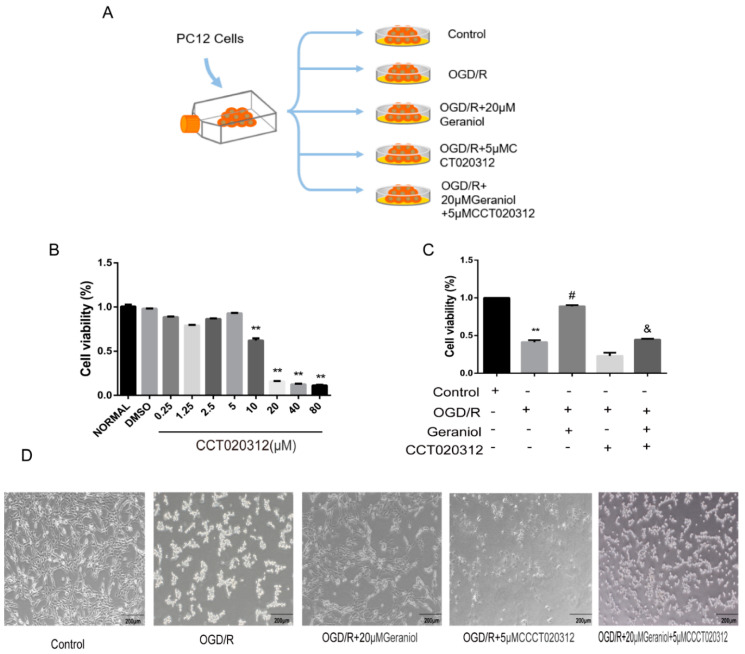
Geraniol had a protective effect on PC12 cells induced by OGD/R through the PERK-ATF4-CHOP signaling pathway in vitro. (**A**) Cell experiment grouping. (**B**,**C**) Cell viability was assayed using a CCK-8 kit (*n* = 3). (**D**) Microscopic observation of the morphology of OGD/R-induced PC12 cells. (**B**) ** *p* < 0.01 CCT020312 group vs. normal group, (**C**) ** *p* < 0.01 OGD/R group vs. control group, # *p* < 0.01 OGD/R+20μM geraniol group vs. OGD/R group, & *p* < 0.01 OGD/R + 20 μM geraniol + 5 μM CCT020312 vs. OGD/R + 20 μM geraniol group.

**Figure 10 ijms-24-00544-f010:**
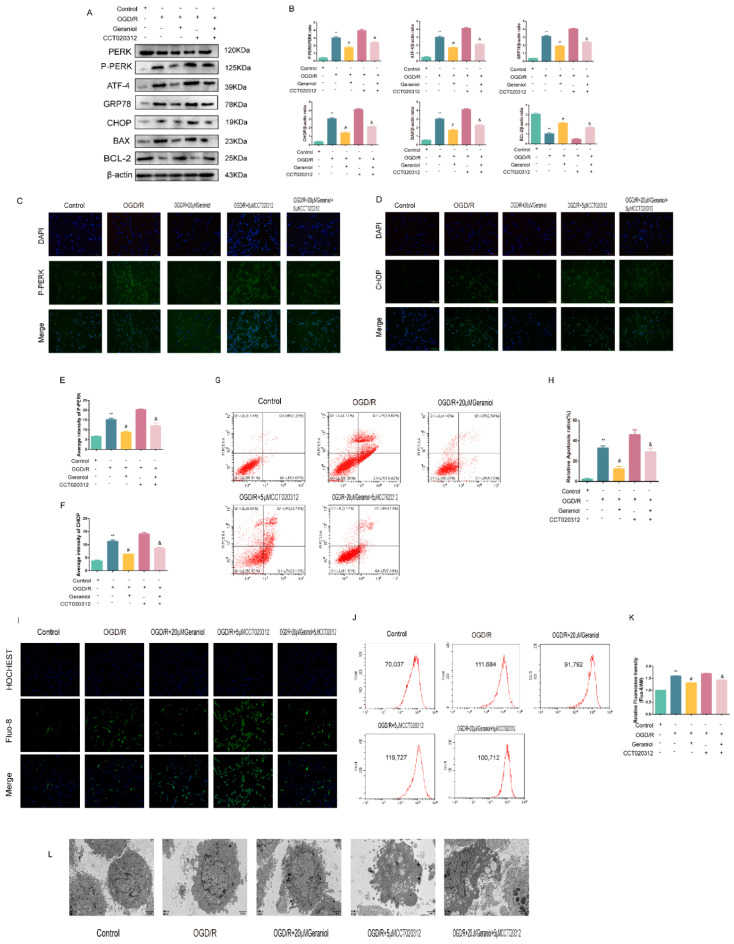
Inhibition of endoplasmic reticulum stress via the PERK-ATF4-CHOP pathway is required for the geraniol-mediated protective effect against oxygen–glucose deprivation/reoxygenation (OGD/R) injury in PC12 cells. (**A**,**B**) Western blot analysis of PERK, P-PERK, ATF4, GRP78, BAX, BCL-2, and CHOP expression (*n* = 6). (**C**–**F**) Immunofluorescence staining of P-PERK and CHOP in the cortex. Representative images were acquired under 400× magnification, scale bars = 50 µm (*n* = 3). (**G**,**H**) Apoptosis detection via flow cytometry (*n* = 3). (**I**) The intracellular calcium ion concentration was detected via Fluo-8/AM staining using laser confocal microscopy. Bar = 200 μm. (**J**,**K**) Flow cytometry detection of intracellular calcium ion concentration via Fluo-8/AM staining (*n* = 3). (**L**) Cell transmission electron microscopy (20,000×). ** *p* < 0.01 OGD/R group vs. control group, # *p* < 0.01 OGD/R + 20 μM geraniol group vs. OGD/R group, & *p* < 0.01 OGD/R + 20 μM geraniol + 5 μM CCT020312 group vs. OGD/R + 20 μM geraniol group.

**Figure 11 ijms-24-00544-f011:**
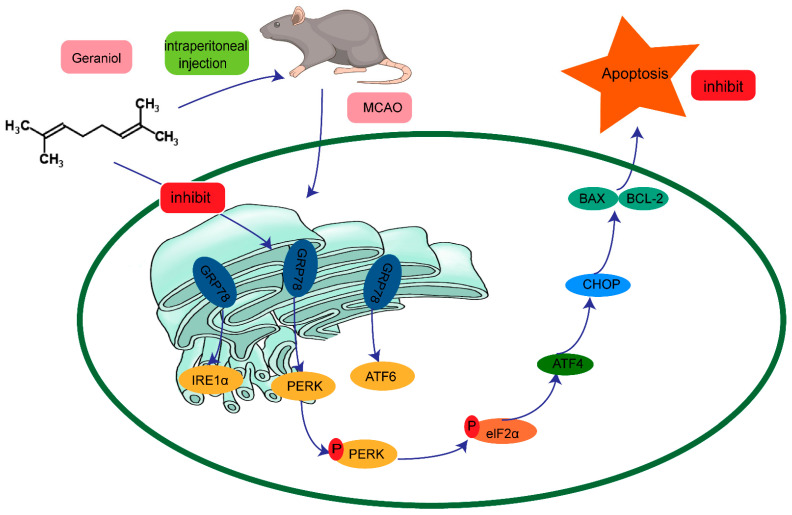
Possible mechanism of the protective effect of geraniol against endoplasmic-reticulum-stress-induced apoptosis after ischemia–reperfusion injury in rat brains.

## Data Availability

All the data are contained within the article.

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
