# Peer review of "Geraniol-Mediated Suppression of Endoplasmic Reticulum Stress Protects against Cerebral Ischemia–Reperfusion Injury via the PERK-ATF4-CHOP Pathway"

_ijms, 2022, doi:10.3390/ijms24010544_

Round 1

Reviewer 1 Report

Geraniol is an acyclic isoprenoid monoterpene isolated from the essential oils of aromatic plants. Geraniol has several biological activities, including antioxidant and anti-inflammatory activity. Recently, geraniol has been demonstrated to have the neuroprotective effect against cerebral I/R injury. The present study demonstrated that the protective effect of geraniol may be related to the endoplasmic reticulum (ER) stress-associated PERK-ATF4-CHOP signaling pathway in a middle cerebral artery occlusion reperfusion model in rats and a PC12 cell oxygen-glucose deprivation/reoxygenation (OGD/R) model. There are some concerns as listed in the following:

(1) Figure 5A is not matched to 4.2. Grouping and drug treatment. [the Geraniol was administered intragastrically to rats once daily for 5 days before the MCAO operation and once daily after the MCAO operation. Lateral ventricular injection of CCT020312 was performed 1.5 hours prior to the MCAO operation.]

(2) In the results of PC12 cell experiments, it is better to show the molar concentration (μM) but not the molar number (μmol) of the tested drugs. [Figures 7-10]

(3) In Figure 7C, it is unclear why geraniol had no significant effects on cell viability at a reoxygenation time of 6 hours and 12 hours but showed a significant effect at 24 hours. It is needed to explain for these results concerning the action of geraniol.

(4) The results of 2.2. Identification of ER stress regulators by Metascape online analysis was not discussed; The results of CCT020312 were not discussed.

(5) Typos and others:

L76: C10H18O

L88: MCAO full name first

L91: dysfunction. (Figure 1A).

L94: manner. (Figure 1B–C).

*L119: protect against CIRI via ER stress?

L138: R group. (Figure 3A–B).

*L146: via ER stress?

Figure 3B: For GRP78 level. I/R+200 #? (significant change?)

L153: ⁎ p<0.01 -> ** p<0.01

**L168: geraniol activated? the PERK-ATF4 168 pathway in the cerebral cortex after I/R

L182: preoperatively. (Figure 5A).

L187: I/R+Geraniol group.(Figure 5E).

L193: I/R+Geraniol group.(Figure 6C–F).

*The title of Figure 5 is the same as that of Figure 6.

*Figure 8D: relative apoptosis cells? -> ratio

**L260: 2.6. Geraniol increases the cellular activity? of OGD/R-induced PC12 cells

L268: OGD/R group. (Figure 7D).

**L231: Geraniol increases the cellular activity? of

L257: CCT020312.(Figure 9A).

L286: OGD/R+Geraniol group.(Figure 286 10I–K).

L293: cytoplasm. (Figure 10L).All

*Figure 9B: Why only 5 μmol group shows ** marker?

**In Figures 9 and 10, it is unclear whether CCT020312 enhances the effects of OGD/R. (no statistical comparison data?)

*The title of Figure 9 is the same as that of Figure 10.

L310: transmission electron microscopy⁎⁎ p<0.01

**L359: Geraniol significantly decreased PERK?, P-PERK, and ATF4 expression

*Figure 11: The target site of Geraniol was not indicated.

**L392: CCT020312 (PERK inhibitor?)

*L435: The source and the passage number of PC12 cells should be indicated.

L470: homogenized with lysate?

L472: ? μg Proteins were loaded

L476: the source of primary antibodies: PERK, P-PERK, ATF-4, GRP78, BAX, BCL-2, CHOP, and β-actin

L483: the source of Cell Counting Kit-8 (CCK-8) solution

L486: the source of Annexin V and 7-AAD

*L492: describe how to calculate apoptosis cells ratio

L495: the source of Fluo-8/AM storage solution and Pluronic F-127 storage solution.

L496: the full name of HHBS

L528-: Refereces: the writing format was not consistent for the title (capital letter, e.g. R2 vs. R3), page number (i.e. R2 vs. R3) [check all]

*L591: R30: S, M. S.; N, M. S.; B, M. M. I.; M, M. E.-S.; S, M. A. E.-H.,?

Reviewer 2 Report

The paper “Geraniol-mediated suppression of endoplasmic reticulum stress protects against cerebral ischemia-reperfusion injury via the PERK-ATF4-CHOP pathway” investigates the protective effect of geraniol against middle cerebral artery occlusion reperfusion in a rat model and against oxygen-glucose deprivation/reoxygenation in a PC12 cell model, with attention to the role of endoplasmic reticulum stress. Different and complementary histochemical and methods and also Western blotting analysis are used in this investigation. Results, their interpretation and conclusions are convincing. However, some faults heavily affect the text, which requires attention and needs to be carefully reconsidered in some parts and revised, as detailed below. Therefore, the decision is rejected, but considering the amount of work done and the results, with a strong recommendation to revise and resubmit.

Some parts of the text need to be revised, to be revised and/or better clarified: For example, as to calcium detection, lines 21 and 497-498, intracellular calcium is detected with the use of the fluorescing calcium binding dye, and the fluorescence of the dye binding calcium, and not of calcium is detected. Hence, “calcium intensity-line 21” and “change in calcium fluorescence intensity-line 498” are wrong, please revise accordingly.

Lines 51-54. There are repetitions, as to lipid and proteins. Please provide only one sentence, including all references, or include text on protein in the first sentence.

Figure 1B,C, Figure5 C,D and lines 420-421 “…Brain infarct volumes were calculated using Image J software”. Since the measurements are performed on tissue sections, there were a procedure to convert measurements on a surface to volumes? Or data relate to surfaces? This does not affect the meaning of the results, but it is to be clarified. Also, the description of the calculation procedure is to be improved, for example by assessing if and how a threshold was defined to distinguish dark and light tissue areas in the images, how the resulting data were defined and normalized to the whole area of the brain slice, and if data refer to pixels or to square/cubic mms of the tissue.  Also, in Materials and methods please specify how tissue was cut, at cryostat, or on ice by a different kind of device?

Lines 110-111 and Table 1. The report on: “59 ER stress regulator genes from published papers (Table 1)” should be completed with related references. These could be reported in Table1, which needs to be implemented with additional information, since the second column is the repetition of the same “Type”. Otherwise, remove it.  Also, Figure 2 presents multiple panels on endoplasmic reticulum (ER) stress regulators, with roles in biological and regulation processes and networks. This is claimed to be derived from a Metascape online analysis. Unfortunately, as provided the panels are poorly informative. In panels A, B, for example, the “how, why and where” the codes have been designed, and the meaning of the x axis values should be better clarified. In panels C-F, it is hard to understand the relationships show by them, also because legends in insets are of bad quality and unreadable. Finally, and more importantly, it is to be clearly stated in the legend if the data processing has been performed by the Authors, or reproduced from literature, in this case, with related references and permission. The 2.2 section is thus to be deeply reconsidered, for example just recalling that in literature and databases many genes/regulators/factors are recognized to be involved in ER signaling, stating then those, and why, have been investigated in this work.

Bright filed images (figures 7, 9) and related text: in Figure 7, from the bars inside pictures, it seems that different magnifications have been used; Figures 7, 9, please standardize bar and character magnification, and correct also for brightness. In the text, please explain the kinds of differences in morphology.  

Section 4.10. For immunofluorescence and DAPI, the observation conditions used at the fluorescence microscope are to be reported, as well as for flow cytometry, as done correctly in section 4.15.

Round 2

Reviewer 1 Report

The present manuscript has revised according to the Editor and Reviewers’ conments. However, the following points or concerns must be rechecked.

#Point 1: Figure 5A is not matched to 4.2. Grouping and drug treatment. -> Figure F5A still is not matched to the drug treatment.

*Figure F5A: (1) Drug treatment only shows 5 days without once after the operation(2) Geraniol was administered intragastrically to rats, but not injection; (3) Lateral ventricular injection of CCT020312 was performed 1.5 hours prior to the MCAO operation, but not injection 1 day prior to surgery.

*4.2. Grouping and drug treatment: the Geraniol was administered intragastrically to rats once daily for 5 days before the MCAO operation and once daily after the MCAO operation vs. Response 1: geraniol was administered continuously 5 days before the operation and once after the operation.

[Point 1: Figure 5A is not matched to 4.2. Grouping and drug treatment. [the Geraniol was administered intragastrically to rats once daily for 5 days before the MCAO operation and once daily after the MCAO operation. Lateral ventricular injection of CCT020312 was performed 1.5 hours prior to the MCAO operation. Response 1: We sincerely thank the reviewer’s comments. The purpose of Figure 5 is to illustrated the time of administration in the animal experiment, and the ischemia time and reperfusion time of the rats in the experiment. Among them, geraniol was administered continuously 5 days before the operation and once after the operation; CCT020312 was administered to the lateral ventricle 1.5h before the operation.]

#Point 2: In the results of PC12 cell experiments, it is better to show the molar concentration (μM) but not the molar number (μmol) of the tested drugs. [Figures 7-10]

*According to the assay volume, it is unclear why original 20 μmol can be changed to 20 μM.

*Figure 7A: cell activity -> cell viability?; geraniol (mol)-> geraniol (M); Add the meaning for markers ** and #.

*Figure 8B: 20 μmol Geraniol -> 20 μM

*Figure 9B: Give the meaning of ** for 5μM CCT020312.

#Point 3: In Figure 7C, it is unclear why geraniol had no significant effects on cell viability at a reoxygenation time of 6 hours and 12 hours but showed a significant effect at 24 hours. It is needed to explain for these results concerning the action of geraniol. Response 3: Thank you for this valuable feedback. It may be related to the drug action time of geraniol. After the cells are induced by hypoxia and hypoglycemia, the number of dead cells increases and the cell viability decreases. However, as the reoxygenation time prolongs, the drug action time also increases, and the cells will Once the activity is restored and reproduction begins, the viability of the cells will be significantly improved.

* It is better to show the cell viability after different reoxygenation time (every OGD/R control vs. OGD/R+ geraniol)

#Point 4: The results of 2.2. Identification of ER stress regulators by Metascape online analysis was not discussed; The results of CCT020312 were not discussed

* Although the purpose and the results of 2.2 were described in the ‘Results’ part, It is better to give some statements to link these results with the present study in the ‘Discussion’ part

* It is better to discuss the use and the results of CCT020312 to confirm the protective effect of Geraniol via inhibiting the PERK-ATF4-CHOP pathway

#Point 5: Typos and others:

L54: CIRI[7, 8] -> CIRI [7, 8]

L60: translocation[11, 12] -> translocation [11, 12]

L62: ER stress[13, 14] -> ER stress [13, 14]

L67: in the cell.[15, 16] -> in the cell [15, 16].

L77: ER stress,the -> ER stress, the

L78: via apoptosis. [16, 17] -> via apoptosis [16, 17].

L86: CHOP[17, 19]

L91: diseases[22, 23]

L92: CIRI[24]

L95: essential oil[25, 26]

L99: apoptosis[28]

L108: in rats. (Figure 1A). -> in rats (Figure 1A).

L112: dose-related(Figure 1B–C)

*L134-136: Gene Ontology and Kyoto Encyclopedia of Genes and Genomes enrichment analyses revealed that these molecules linked to ER stress,such as Eif2a. Atf4Atf6BCL2Hsp90aa1Bax and Creb3. -> Gene Ontology and Kyoto Encyclopedia of Genes and Genomes enrichment analyses revealed that these molecules linked to ER stress, such as Eif2a, Atf4, Atf6, BCL2, Hsp90aa1, Bax and Creb3.

*L163: attenuate apoptosis after I/R 163 in rats via ER stress? -> via inhibiting ER

L170: I/R group(Figure 3A–B)

L227: Geraniol group.(Figure 5E). -> Geraniol group (Figure 5E).

L234: Geraniol group.(Figure 6C–F) -> Geraniol group (Figure 6C–F).

L237: I/R+Geraniol+CCT020312 group.(Figure 6G–H). -> I/R+Geraniol+CCT020312 group (Figure 6G–H).

L259: (A–B)Western; L260: (C–F)Immunofluores; L262: .(C–F)Immunofluores

L272: 20 μmol geraniol -> change to concentration

L279: intercellular joints. (Figure 7D). -> intercellular joints (Figure 7D).

L302: the OGD/R group. (Figure 8C–D). -> the OGD/R group (Figure 8C–D).

L306: Figure 8D: the label ‘Relative Apopotosis cells’ -> Relative Apopotosis Cells (%)

L315: CCT020312.(Figure 9A) -> CCT020312 (Figure 9A)

L320: treatment. (Figure 9B–C). -> treatment (Figure 9B–C).

L336: Geraniol group. (Figure 10G–H). -> Geraniol group (Figure 10G–H).

L347: group.(Figure 10I–K). -> group (Figure 10I–K).

L353: Cells in -> cells in

L354: shrinkage. (Figure 10L). -> shrinkage (Figure 10L).

L359: Figure 9A: the label 20 μmol -> 20 μM; 5 μmol -> 5 μM

*L432: Geraniol significantly decreased PERK? -> Check the data in Figures 6 A,B and 10A,B, no bar figure was shown for PERK only and the PERK protein expression seems to be not significantly changed in each group (Figure 6A ad Figure 10A)?

*L437: pathway.( (Figure 11).

*Figure 11: The reperfusion should be showed; the injection needle should be changed to an intragastrical tube; the inhibition sign or marker should be added for Geraniol; and Apoptosis : L452: give the source of the rats

L466: CCT020312(PERK Activator,

L468: Lateral ventricular injection-> give injection volume

L497: dema.Infarct

L502: paraformaldehyde(Beyotime, P0099)

L512: ATCC(American Type

*L514 add: The third passage of PC12 cells was used for experiments.

L516: serum(FBS,

L517: streptomycin(Beyotime,

L538: DAPI(Beyotime,

L541: Observe at 400 times the field of view The images

L552: buffer(Be-

L557: gels(Epizyme

L559: Tween 20(Beyotime

*L561: PERK(Dilution:1:1000,Affinity Bioscience,AF5304), P-PERK(Dilu-  tion:1:1000,Affinity Bioscience,AF4499), ATF-4(Dilution:1:800,Affinity Biosci-  ence,DF6008), GRP78(Dilution:1:1000, Bioss,bs-1219R), BAX(Dilution:1:1000,Affinity Bio-science,AF0120), BCL-2(Dilution:1:1000,Affinity Bioscience,AF6139),CHOP(Dilu- tion:1:1000,Affinity Bioscience,AF6277), and β-actin(Dilution:1:1000,Proteintech, 565 10021293.). -> leave one space between words

L569: Proteintech,B900210

L573: CCK-8,Beyotime,C0037

L577: 7-AAD(Solar-bio,CA1030)

L586: Fluo-8/AM(AAT-Bioquest,1345980-40-6)

L588: HHBS(Hanks'Buffer with 20 mM Hepes, AATBioquest,AAT-

*L625: References: the writing format was not consistent for the title (e.g. R3) and page number (e.g. R2) [check all again]

R2: 181-98 -> 189-198

R3: Ischemia-Reperfusion Injury After… -> Ischemia-reperfusion injury after…

R4, R5, R6, R8, R10, R12, R13, R17, R18, R19, R20, R21, R22, R25, R27, R28, R30, R31, R34, R35, R38, R40, R41, R43, R44, R45, R47  

*R31: S, M. S.; N, M. S.; B, M. M. I.; M, M. E.-S.; S, M. A. E.-H.,? -> Kim, S. H.; Bae, H. C.; Park, E. J.; Lee, C. R.; Kim, B. J.; Lee, S.; Park, H. H.; Kim, S. J.; So, I.; Kim, T. W.; Jeon, J. H., Geraniol inhibits prostate cancer growth by targeting cell cycle and apoptosis pathways. Biochem Biophys Res Commun 2011, 407, (1), 129-134.

Reviewer 2 Report

Major remarks:

Lines 51-54. Repetitions, as to lipid and proteins, have not been revised, as previously suggested.

Figure 1B,C, Figure5 C,D and lines 420-421 “…Brain infarct volumes were calculated using Image J software”: The question on volume or area is still unsolved (line 495: “Image J was used to evaluate the infarction area of each slice. The infarct volume…..”). In other words, it is to state clearly if data (figures 1C,5D) are referred to areas, as it seems from the anyway insufficient description in “Materials and Methods”, and if data are as pixels or to square/mms of the tissue.  Following this mandatory clarification, corrections on Volume/ area are to be given along the text everywhere this is pertinent, including Figure axes and legends. Also, the device, or a, blade used to cut brain tissue slices on ice has not been indicated.

Table 1. The sentence al line 133 still refers to Table 1, which has been removed from the text and now provided as Table S1, as indicated at line 606, and anyway still missing references and/or WEB site as the sources of information.

Also Figure 2 has not been duly revised. As to the quality of presentation, some insets have been doubled with a higher magnification, but keeping their less magnified version makes the figures to be still of poor quality. As to the source and meaning of data, explanations given in the cover letter response should be also to be provided in the paper.

Minor remark:

Line 592 – “by confocal laser”, should be “by confocal laser microscopy”, is it?

Bright field images (figures 7, 9) the bars appear now to have been standardized. However, characters are poorly readable, and the pictures of the cells seem unchanged, contrarily to the extent of bars, which before were different (please see the picture in the middle vs the others) and now are almost the same.

As to Section 4.10, and observation of immunofluorescence and DAPI, besides the model of the microscope used, the conditions to be reported are excitation and emission wavelengths, as done correctly at lines 594-595, indicated as to example in the former revision remark.

Round 3

Reviewer 1 Report

The present manuscript has revised according to the Editor and Reviewers’ comments. However, the following points or concerns must be rechecked before it can be accepted for publication.

*(1) Figure 7A can be deleted, because Figure 7B already shows the effects of various concentrations of Geraniol on normal cell viability. It appears that Geraniol has no significant effect on cell viability from 5-320 μM (NS, not only ** on 20 μM), which contradict to those of 1x10-5 and 1x10-4M in Figure 7A.  

*(2) Figure 8B, the ** should be labelled on 10, 20, 40, 80 μM, but not 5 μM. (L386)

(3) Typos and others

L17: is unclear.Purpose:The aim

*L78: while favoring the upregulation. ER Stress ?

L118: was dose- related -> was dose-related

*L219-220: and unclear or absent ER compared with the I/R group. The I/R+200mg/kg Geraniol group -> and unclear or absent ER. Compared with the I/R group, the I/R+200mg/kg Geraniol group

*L272: & p<0.01 I/R+5mmol CCT020312 vs I/R+200mg/kg Geraniol group -> & p<0.01 I/R+200mg/kg Geraniol+5mmol CCT020312 vs I/R+200mg/kg Geraniol group

*L285-286: & p<0.01 I/R+5mmol CCT020312 vs I/R+200mg/kg Geraniol group -> & p<0.01 I/R+200mg/kg Geraniol+5mmol CCT020312 vs I/R+200mg/kg Geraniol group

*L290: we first induced PC12 cells -> we first treated PC12 cells

L296: 24h.And

*L309-310: delete Figure 7A; no any ** on the Figure 7B; only label ** on OGD/R+20 μM Geraniol group instead of #, *, **

*L313-316: (A-C) -> (A-B); (D)-> (C); delete (A) ⁎⁎ p<0.01 10^6 Geraniol group vs Normal group, (B) ⁎⁎ p<0.01 20 μM Geraniol group vs Normal group; (C)->(B) ⁎⁎ p<0.01 OGD/R+20 μM Geraniol group vs OGD/R control group

L336: Bcl-2 vs. BCL-2 -> consistently use one in the text.

L337: OGD/R group vs sham group -> OGD/R group vs Control group

L350: OGD/R+CCTO20312+Geraniol -> OGD/R+ Geraniol+CCTO20312 (L345)

L355: CCTO20312+Geraniol group -> OGD/R+Geraniol+CCTO20312 group

*L386: Figure 9A: ** should be labelled on 10, 20, 40, 80 μM, but not on 5 μM; Figure 9C: Sham -> Control

*L392: (B)-> ** p<0.01 5μM CCT020312 group vs Normal><0.01 CCT020312 group vs Normal group [ ** should be labelled on 10, 20, 40, 80 μM, but not 5 μM]

*L394: -> OGD/R+ Geraniol+CCT020312 vs OGD/R+20μM Geraniol group

*L409: OGD/R group vs sham group -> OGD/R group vs Control group; OGD/R+CCT020312 group vs OGD/R+20μM Geraniol group -> OGD/R+ Geraniol +CCT020312 group vs OGD/R+20μM Geraniol group

*L431-432: And according to our online analysis of 59 ER stress-related molecules using Metascape (https://metascape.org/gp/index). The analysis showed that.. -> According to our online analysis of 59 ER stress-related molecules using Metascape (https://metascape.org/gp/index), the results showed that…

L506: Lateral ventricular injection of 5mmol CCT020312 -> give the injection volume (?μl)

L558: USA).The third; for experiments.PC12

L623: 4.14. Flow cytometric apoptosis assays: give how to define the apoptosis cell ratio.  

L675-: Reference: check the capital letter in the title

R3 (L682): after Endovascular Thrombectomy for Ischemic Stroke

R4: in Cerebral Ischemia Reperfusion Injury.

R5: for Cerebral Ischemia Reperfusion Injury.

R11: The Unfolded Protein Response in …

R15: Icariin Inhibits Endoplasmic Reticulum …

R20: Cerebral Ischemia by inhibiting Endoplasmic Reticulum …

R21: exacerbates Ischemic Stroke following …

R22: Roles of Endoplasmic Reticulum Stress in immune responses

R30: Cerebral Ischemia/Reperfusion-Induced Injury …

R31: targeting Cerebral insult a

R38: Ischemic Stroke-induced neuronal …

R45: Endoplasmic Reticulum …

R49: CCT020312 inhibits Triple-Negative Breast cancer …

Reviewer 2 Report

The paper “Geraniol-mediated suppression of endoplasmic reticulum stress protects against cerebral ischemia-reperfusion injury via the PERK-ATF4-CHOP pathway” has been now duly revised according to the Reviewer’s remarks, apart two remaining questions.

One is about the quality of image 2, still not full satisfying. In this case I think that Editorial staff will be able to manage the question with Authors to get an improvement.  

The presentation of data on brain infarct is still unspecified. I consider that the correctness of data presentation is essential for the quality of the Journal. The use of “area” or “volume” to identify the results and the clarification on the measuring units are mandatory. So, a further effort is asked to report along all the text on “area” or “volume”, likely “area” considering the description of the measuring procedure performed on images by means of ImageJ. Also, it is mandatory to define accordingly the measuring units used, as square millimeters, or microns, or pixels, or as cubic millimeters, or microns, or pixels, in all related figures and parts of the text. This is the reason for the request of major revision.

Reviewer 3 Report

The authors provided most of the queries. The concern about the western still there. 

The cut blots do not prove the right molecular weight. If a prestained ladder image is there and some marking that supports the appropriateness, then, it will be fine. That is why major revisions are asked earlier to re-run the blots.

Round 4

Reviewer 2 Report

The paper “Geraniol-mediated suppression of endoplasmic reticulum stress protects against cerebral ischemia-reperfusion injury via the PERK-ATF4-CHOP pathway” has been duly revised according to the Reviewer’s remarks, and is now suitable for publication in IJMS.

Author Response

Thanks for your suggestion to our article

Reviewer 3 Report

There is no change in the asked concern.

Sending the same blots in different orientations does not really change anything. 

My suggestion is to do it in the right way.